# Analysis of the Association of Two SNPs in the Promoter Regions of the *PPP2R5C* and *SLC39A5* Genes with Litter Size in Yunshang Black Goats

**DOI:** 10.3390/ani12202801

**Published:** 2022-10-17

**Authors:** Peng Wang, Wentao Li, Ziyi Liu, Xiaoyun He, Rong Lan, Yufang Liu, Mingxing Chu

**Affiliations:** 1Key Laboratory of Animal Genetics, Breeding and Reproduction of Ministry of Agriculture and Rural Affairs, Institute of Animal Science, Chinese Academy of Agricultural Sciences, Beijing 100193, China; 2Yunnan Animal Science and Veterinary Institute, Kunming 650224, China

**Keywords:** goat, litter size, *PPP2R5C*, *SLC39A5*, single-nucleotide polymorphism (SNP)

## Abstract

**Simple Summary:**

The protein phosphatase 2 regulatory subunit B’gamma (*PPP2R5C*) and solute carrier family 39 member 5 (*SLC39A5*) genes are essential in mammalian growth and development. Here, KASP genotyping was used to analyze the association between polymorphisms of *PPP2R5C* and *SLC39A5* and litter size in Yunshang black goats, a cultivated goat breed with high prolificacy in China. The results show that *PPP2R5C* g.65977743C>T and *SLC39A5* g.50676693T>C were significantly associated with litter size of the third parity in Yunshang black goats. Luciferase assays and RT-qPCR showed that individuals with the CC genotype of *PPP2R5C* and *TT* genotype of *SLC39A5* showed higher expression of these genes in goat ovarian tissues than those with other genotypes. Transcription factor predictions showed that *SOX18*, *ZNF418*, and *ZNF667* and *NKX2-4* and *TBX6* bind to the polymorphisms of two respective genes. These two polymorphisms provide new insights into the study of fertility in goats.

**Abstract:**

Screening for candidate genes and genetic variants associated with litter size is important for goat breeding. The aim of this study was to analyze the relationship between single nucleotide polymorphisms (SNPs) in *PPP2R5C* and *SLC39A5* and litter size in Yunshang black goats. KASP genotyping was used to detect the SNP genetic markers in the *PPP2R5C* and *SLC39A5* in a population of 569 Yunshang black goats. The results show that there were two SNPs in the *PPP2R5C* and *SLC39A5* promoter regions. Association analysis revealed that the polymorphisms *PPP2R5C* g.65977743C>T and *SLC39A5* g.50676693T>C were significantly associated with the litter size of the third parity of Yunshang black goats (*p* < 0.05). To further explore the regulatory mechanism of the two genes, the expression of different genotypes of *PPP2R5C* and *SLC39A5* was validated by RT-qPCR and Western blotting. The expression of *PPP2R5C* was significantly higher in individuals with the *TT* genotype than in those with the *TC* and *CC* genotypes (*p* < 0.05). The expression of *SLC39A5* was also significantly higher in individuals with the *TT* genotype than in *TC* and *CC* genotypes (*p* < 0.05). Dual luciferase reporter analysis showed that the luciferase activity of *PPP2R5C*-C variant was significantly higher than that of *PPP2R5C*-T variant (*p* < 0.05). The luciferase activity of *SLC39A5*-T variant was significantly higher than that of *SLC39A5*-C variant (*p* < 0.05). Software was used to predict the binding of transcription factors to the polymorphic sites, and the results show that *SOX18*, *ZNF418*, and *ZNF667* and *NKX2-4* and *TBX6* might bind to *PPP2R5C* g.65977743C>T and *SLC39A5* g.50676693T>C, respectively. These results provide new insights into the identification of candidate genes for marker-assisted selection (MAS) in goats.

## 1. Introduction

Goats were one of the first livestock species to be domesticated [1]. The domestication of goats was crucial to the development of agriculture and has played an important role in the development of human society [2,3]. Although goat breeding resources are abundant, low fertility and slow growth rates have hindered the rapid development of the goat industry [4]. Therefore, it is important to identify the molecular markers that have an impact on the litter size of goats.

Protein phosphatase 2 regulatory subunit B’gamma (*PPP2R5C*) is a subunit of protein phosphatase 2A (PP2A), a major protein serine/threonine phosphatase, and its encoding gene is located on goat chromosome 21 with a CDS region consisting of 14 exons [5]. PP2A plays an important regulatory role in protein phosphorylation and dephosphorylation and is involved in the regulation of cell proliferation, differentiation, migration, and numerous signal transduction pathways [6,7,8]. PP2A has both positive and negative regulatory effects on the cell cycle [9]. *PPP2R5C* is one of the subunits of PP2A and alterations or polymorphisms in its expression pattern result in altered protein phosphorylation. In human lung cancer cell lines, a polymorphism of *PPP2R5C* at the 14q32.31 locus resulted in blocked dephosphorylation of any substrate and the inhibition of tumor cell growth [10]. In human overgrowth studies, the presence of five variant loci was found to potentially affect substrate binding, thereby interfering with the ability of PP2A to dephosphorylate specific protein substrates, causing overgrowth in humans [5]. In the liver of *PPP2R5C* knockout mice, systemic glucose tolerance was found to be produced and insulin sensitivity was improved, leading to altered protein kinase AMP-activated catalytic subunit alpha 1 (*AMPK*) and sterol regulatory element binding protein 1 (*SREBP-1*) activity, which in turn affected insulin synthesis [11]. Solute carrier family 39 (metal ion transporter), member 5 (*SLC39A5*) is located on goat chromosome 5, consists of 10 exons and is one of the 14 members of the SLC39 zinc transporter family [12]. Most of SLC39’s proteins are located on the cell membrane, and some are located on organelles [13]. SLC39 proteins are important transporter proteins for the transport of metal ions from extracellular or intracellular organelles into the cytoplasm [14]. *SLC39A5*, one of the key components of SLC39 protein, also plays an important role in the transport of metal ions such as zinc ions across the cell membrane [15]. *SLC39A5* has been shown to play an important role in Zn^+^ transport within intestinal epithelial cells in vivo, maintaining systemic homeostatic control [16]. In addition, *SLC39A5* plays an important role in physiological and pathological conditions [17]. For example, knockout of the mouse beta cell specific *SLC39A5* resulted in impaired insulin secretion and reduced glucose tolerance [18]. High levels of *SLC39A5* expression in zebrafish venous vessels regulate zinc homeostasis and play a key role in endothelial sprouting and migration in venous angiogenesis [19]. The above findings suggested that *PPP2R5C* and *SLC39A5* play a very important role in the growth and development of animals.

The candidate genes in this study were derived from our group’s previous resequencing data of Yunshang black goats, and we found that *PPP2R5C* and *SLC39A5* were associated with litter size in goats. These genes are related to cell proliferation and regulation. KASP genotyping utilizes a unique form of competitive allele specific PCR combined with a fluorescence-based reporter system for the identification and measurement of genetic variation occurring at the nucleotide level to detect single nucleotide polymorphisms (SNPs) or insertion fragments and deletions (Indels) [20]. In this study, KASP genotyping technology was used to detect SNP genetic markers in the promoter regions of the *PPP2R5C* and *SLC39A5* in the Yunshang black goat population [20]. Subsequently, RT-qPCR and Western blotting were used to verify the differential expression of *PPP2R5C* and *SLC39A5* in individuals with different genotypes. A dual luciferase reporter assay detected the activity of the promoter region containing the polymorphisms and predicted the possible binding of transcription factors. Our results show that *PPP2R5C* g.65977743C>T and *SLC39A5* g.50676693T>C were significantly associated with the number of lambs born in Yunshang black goats, and we were able to identify the expression levels of *PPP2R5C* and *SLC39A5* in individuals with different genotypes and predicted transcription factors. These results are expected to provide additional information for the development of molecular breeding of goats.

## 2. Materials and Methods

### 2.1. Ethics Statement

All animals were authorized by the Department of Scientific Research (responsible for animal welfare), Institute of Animal Science, Chinese Academy of Agricultural Sciences (IAS-CAAS; Beijing, China). In addition, the IAS-CAAS Animal Ethics Committee approved the ethics of animal survival (No. IAS2021-25).

### 2.2. Experimental Samples

All goats were raised in Honghe Hani and Yi Autonomous Prefecture, Yunnan Province, China. This farm adopts the method of natural random mating in the goat population. The female goats were of similar age and weight and were reared and managed in the same environment. Blood samples were collected from 569 Yunshang black goats (ages 2–5 years old) for which at least 1 litter was recorded (Table 1). Ten ml of blood was collected through a vein for each sample. The blood was anticoagulated with K2EDTA anticoagulant and stored at −20 °C. Afterward, 300 µL was taken to extract genomic DNA. The female goats with different *PPP2R5C* and *SLC39A5* genotypes were selected based on KASP genotyping, and ovarian tissues were collected after slaughter, placed in liquid nitrogen and stored at −80 °C. Three individuals per group. The individuals used for sampling were randomly selected from the goat farm, and there was no direct relationship among them.

### 2.3. DNA and RNA Extraction, and Preparation of cDNA

Genomic DNA was extracted from the 569 goats blood samples according to the instructions of the Animal Blood DNA Extraction Kit (Tiangen, Beijing, China). The DNA was dissolved in enzyme-free sterile water (DNase/RNase-free ddH_2_O) (Tiangen, Beijing, China), the DNA sample concentration was measured with a NanoDrop 2000 (Thermo, Waltham, MA, USA), and a 1.2% agarose gel was used to detect DNA quality. The extracted DNA concentrations ranged from 300 to 500 ng/μL and were then diluted to 50 ng/μL for subsequent experiments.

Total RNA extraction was performed according to the instructions of the TRIzol kit (Tiangen, Beijing, China). RNA sample concentration and quality were detected with a NanoDrop 2000 (Thermo, Waltham, MA, USA) and 1.2% agarose gels. Then, the RNA was reverse transcribed into cDNA using a reverse transcription kit (TaKaRa, Kusatsu, Japan) and stored at −20 °C.

### 2.4. Primer Design and Amplification for SNPs

To detect SNPs in the promoter regions of *PPP2R5C* and *SLC39A5*, primers were designed with Primer v5 software to amplify the first 2000 bp of the prokinetic regions of these two genes, using sequences from the Ensembl (https://asia.ensembl.org/index.html, accessed on 22 May 2022) database as a reference. PCR products from 40 selected Yunshang black goats were sent to Shanghai Biotechnology Service Co. The Sanger sequencing results were analyzed by Dnastar6.0 (Dnastar, Madison, WI, USA) software. Information on polymorphic loci of *PPP2R5C* and *SLC39A5* were determined, and then 569 collected Yunshang black goats were genotyped for KASP. Information on polymorphic loci of *PPP2R5C* and *SLC39A5* was determined, and the 569 collected Yunshang black goat samples were genotyped using KASP. The primers were synthesized by the Shanghai Biotechnology Service Co. The PCR amplification system included 1 μL of DNA template, 10 μL of 2 × PCR Master Mix, and 0.51 μL each of 10 μmol/L forward and reverse primers. PCR amplification procedure: 94 °C predenaturation for 3 min; denaturation at 94 °C for 30 s, annealing at 59 °C for 30 s, extension at 72 °C for 30 s, 35 cycles; final extension at 72 °C for 5 min. PCR products were detected using 1.2% agarose gels and stored at 4 °C. The primers for different fragments are shown in Table 2.

### 2.5. Identification and Genotyping of Single Nucleotide Polymorphisms (SNPs)

In 569 Yunshang black goats *PPP2R5C* and *SLC39A5* SNP loci were detected using the KASP typing technique. KASP typing technique provided by the Tianjin Compsen Biotechnology Co. Information on the relevant primers is shown in Table 3. The samples used for genotyping were previously extracted genomic DNA, with each requiring a volume of 20 μL with a DNA concentration of 40–80 ng/μL.

### 2.6. RT-qPCR Validation

RT-qPCR was performed using SYBR Green qPCR Mix (TaKaRa, Dalian, China) and RocheLight Cycler^®^ 480 II system (Roche Applied Science, Mannheim, Germany) according to the manufacturer’s protocol. RT-qPCR was performed in a volume of 20 µL containing approximately 0.8 µL of 10 µmol/L of each primer, 10 µL of SYBR Green qPCR mix, 2.0 µL of 50 ng/µL of genomic DNA, and the rest in ddH_2_O. RT-qPCR conditions were as follows: denaturation at 95 °C for 5 min, followed by a period of 5 min in the Roche LightCycler^®^480 II system (Roche Applied Science, Mannheim, Germany) for 40 cycles, denaturing at 95 °C for 5 s and annealing at 60 °C for 30 s. The goat *RPR19* was used as a reference gene. The information of RT-qPCR primer sequences is shown in Table 4. The primers were synthesized at Shanghai Biotechnology Co., Ltd.

### 2.7. Protein Extraction and Western Blotting

Goat ovarian tissues were incubated on ice for 30 min in 1.5 mL centrifuge tubes containing lysis buffer, and then centrifuged at 12,000× *g* for 10 min at 4 °C. The supernatant was collected and stored at −80 °C. The protein concentration was measured with a BCA assay kit (Solarbio, Beijing, China), and then 30 μg of protein was mixed with protein loading buffer (Solarbio, Beijing, China) and boiled at 99 °C for 10 min. The protein samples were then subjected to gel electrophoresis at 100 V on a 10% Bis-Tris gel. The proteins were transferred onto a nitrocellulose membrane and blocked at 4 °C overnight. Anti-PPP2R5C and anti-SLC39A5 were purchased from Proteintech company (Proteintech, Wuhan, China). All primary antibodies were diluted at 1:10,000 with primary antibody diluent (P0256-500 mL, Bain-marie, China) and washed three times with TBST before being washed with HRP-conjugated anti-rabbit IgG (Proteintech, Wuhan, China) with secondary antibody dilution (P0258-500 mL, Baymax, China) diluted to 1:5000 used to incubate the membrane for 2 h at room temperature. Western blots were visualized on an Odyssey CLX imaging system (Li-COR) (Bio-Rad, Hercules, CA, USA) with Supersignal HRP chemiluminescent substrate (Beyotime, Beijing, China) according to the manufacturer’s instructions.

### 2.8. Bioinformatics Analysis

JASPAR (http://jaspar.genereg.net, accessed on 23 May 2022) online software was used to predict the transcription factors that might bind to the *PPP2R5C* g.65977743C>T and *SLC39A5* g.50676693T>C polymorphic sites.

### 2.9. Plasmid Construction and Dual Luciferase Assay

To assess the promoter activity of the different genotypes, the plasmids of wild type and variant *PPP2R5C* and *SLC39A5* (named *PPP2R5C*-T variant and *PPP2R5C-*C variant, and *SLC39A5*-C variant and *SLC39A5*-T variant, respectively) were constructed. A sequence including +/−100 bp before and after the mutant site was inserted into the luciferase vector pGL3-basic and the promoter activity of the fragment was verified by cell transfection.

Cell transfection was performed in 24-well plates reached Lipofectamine 2000 (Thermo Fisher, Waltham, MA, USA). Transfection was performed when the density of HEK293T cells in the 24-well reached approximately 70%. Transfections were performed using Opti-DMEM (Thermo Fisher, USA). The medium was replaced 6 h after transfection, and cells were collected after 48 h and transferred to 96-well plates for the dual luciferase activity assay. Firefly and Renilla luciferase activities were measured using a dual luciferase reporter assay kit (Promega, Beijing, China) and a Modulus single tube multimode reader (Turner Biosystems, Sunnyvale, CA, USA) according to the manufacturers’ protocols.

### 2.10. Statistical Analysis

The observed genotypes, allele frequencies, Hardy–Weinberg equilibrium (HWE), and population indices (polymorphism information content, PIC; homozygosity, He; number of effective alleles, Ne) were calculated using PopGenev 1.31 (http://www.ualberta.ca/∼fyeh/fyeh, accessed on 23 May 2022). The chi-square test was used to determine whether the distribution of each genotype deviated from Hardy–Weinberg equilibrium [21].

The association between SNPs and number of litter sizes born was analyzed using the general linear equation formula.
y_ij_ = μ + HYS_i_ + G_j_ + e_ij_
where, y_ij_ is the observed litter size, μ is the population mean, HYS_i_ is the fixed effect of the herd-year-season, G_j_ is the fixed effect of genotype or diplotype, and e_ij_ is the random error. The litter sizes of the first three parities were used in this study.

One-way ANOVA was used to test the hypothesis that several means were equal. Statistical analysis was performed on the collected data by using SPSS 21.0 (SPSS Inc., Chicago, IL, USA) statistical software, and the average of three replicates was evaluated and displayed as the mean ± standard error (SE). Student’s t test was used to test for significant differences in the data between groups. * *p* < 0.05 and ** *p* < 0.01 were considered significant differences.

## 3. Results

### 3.1. Identification and Typing of PPP2R5C and SLC39A5 SNPs

Sanger sequencing could identify two SNPs, PPP2R5C g.65977743C>T and SLC39A5 g.50676693T>C. However, the KASP method was used to genotype the SNPs (Figure 1A, B). The results show that there were 131 individuals with *TT* genotype, 324individuals with *TC* genotype and 124 individuals with *CC* genotype on the *PPP2R5C* g.65977743C>T. There were 139 individuals with *TT* genotype, 257 individuals with *TC* genotype and 173individuals with *CC* genotype on the *SLC39A5* g.50676693T>C polymorphism. Both of these loci were reported in Ensembl.

### 3.2. Polymorphism Analysis of PPP2R5C and SLC39A5

Genotype frequencies of *PPP2R5C* g.65977743C>T and *SLC39A5* g.50676693T>C were calculated for 569 Yunshang black goats, He, Ne and PIC (Table 5). Three genotypes were found for SNP g.65977743C>T. *CC* was the most common genotype with a genotype frequency of 0.42, which was higher than *CT* (0.39) and *TT* (0.19). The most common allele was T with a higher allele frequency of 0.62 than C (0.39). χ^2^ value was 0.01, confirming that the frequency distribution of SNP g.65977743C>T in the selected black goat population on clouds was in accordance with the Hardy–Weinberg equilibrium law. Among SNP g.50676693T>C, *TC* was found to be the most common genotype with a genotype frequency of 0.45, which was higher than *CC* (0.23) and *TT* (0.31). The most common allele was T with a higher allele frequency of 0.54 than C (0.46). χ^2^ value of 0.03 confirmed that the frequency distribution of SNP g.65977743C>T and g.50676693T>C in Yunshang black goat was in accordance with the Hardy–Weinberg equilibrium law.

### 3.3. Analysis of the Associations between PPP2R5C g.65977743C>T and SLC39A5 g.50676693T>C and the Litter Size in Yunshang Black Goats

Descriptive statistics of litter size phenotypes are shown in Table 6. With respect to the *PPP2R5C* g.65977743C>T polymorphic locus, *TT* individuals had a higher number of lambs in the first three litters than did *TC* and *CC* individuals. However, both the *TT* and *TC* genotypes were associated with a significantly higher number of lambs (*p* < 0.05) than the *CC* genotype in the third lambing. For the *SLC39A5* g.50676693T>C polymorphic locus, *TT* individuals had a higher number of lambs than did individuals with the *TC* and *CC* genotypes in the first three parities. The number of lambs in the third litter was significantly higher for the *TT* than for the *TC* and *CC* genotypes (*p* < 0.05) (Table 6).

### 3.4. Expression of PPP2R5C and SLC39A5 with Different Genotypes in Goat Ovarian Tissues

To verify the expression of *PPP2R5C* and *SLC39A5* in individuals with different genotypes in goat ovarian tissues, RT-qPCR and Western blotting were performed. The RT-qPCR results show that the expression of *PPP2R5C* in the individuals with *TT* genotype was significantly higher than that in the individuals *TC* and *CC* genotype (*p* < 0.05) (Figure 2A). The expression of *SLC39A5* in the individuals with the *TT* genotype was significantly higher than that in individuals with the *TC* and *CC* genotypes (*p* < 0.05) (Figure 2A). The western blotting results are consistent with the results of RT-qPCR (Figure 2B).

### 3.5. SNPs Affect on the Promoter Activity of PPP2R5C and SLC39A5

To further investigate the effects of SNPs on the regulation of gene transcription, the transcriptional activity of the promoter regions of *PPP2R5C* and *SLC39A5* with different genotypes was examined using a dual luciferase assay. The results show that the transcriptional activity of the promoter region of the plasmid with the *TT* genotype was higher than that of the plasmid with the *CC* genotype in *PPP2R5C* (*p* < 0.05) (Figure 3A). In the study of the transcriptional activity of the *SLC39A5* promoter region, the plasmid with the *TT* genotype showed higher activity than the plasmid with the *CC* genotype (*p* < 0.01) (Figure 3B).

### 3.6. Prediction of Transcription Factor binding Sites Affected by the PPP2R5C and SLC39A5 Polymorphisms

Potential transcription factors binding sites in the promoter sequences where the *PPP2R5C* g.65977743C>T and *SLC39A5* g.50676693T>C polymorphic loci were predicted by the online software JASPAR (http://jaspar.genereg.net, accessed on 23 May 2022) [22]. When the *PPP2R5C* g.65977743C>T locus was T, no transcription factor was predicted to bind to it. When the locus was C, transcription factors *SOX18*, *ZNF418* and *ZNF667* were predicted to bind to it (Figure 4A). When the *SLC39A5* g.50676693T>C site was C, the transcription factor *NKX2-2* was predicted to bind to it. When the site was T, *NKX2-2*, *NKX2-4* and *TBX6* were predicted to bind to it (Figure 4B). Both genes produced mutations that resulted in an increase in the transcription factor.

## 4. Discussion

Recently, single nucleotide polymorphisms (SNPs) have been used for the study of reproduction in goats [23]. In a study on Liaoning velvet goats, Chang et al. found that the C allele of the bone morphogenetic protein receptor type 1B (*FecB*) C94T and estrogen receptor 1(*ESR*) C463T loci were significantly associated with the litter size produced by LCG [24]. Hui et al. found that an 11 bp insertion deletion in DNA methyltransferase 3 beta (*DNMT3B*) was significantly associated with the number of first lambs born in goats [25]. Litter size is an important factor affecting the economic efficiency of goats. The candidate genes in this study were derived from our group’s previous resequencing data of Yunshang black goats, and we found that *PPP2R5C* and *SLC39A5* were associated with litter size of goats [26]. *PPP2R5C* and *SLC39A5* play regulatory roles in the PI3K-AKT pathway and mitochondrial metabolism, respectively [27,28]. Numerous studies have shown that the PI3K-AKT pathway and mitochondrial metabolism have important effects on mammalian reproduction [29,30,31]. Therefore, we examined the effect of SNPs in the promoter regions of *PPP2R5C* and *SLC39A5* on lambing numbers in Yunshang black goats on clouds.

The promoter region and the binding region of RNA polymerase are directly related to the efficiency of transcription, influence the expression of genes, and regulate gene function [32]. Studies have shown that not only limited to coding regions, but also genetic variants in noncoding regions can affect gene action. [33]. In mammals, SNPs polymorphism in the promoter region were found to be central to the transcriptional regulatory mechanism in vitro, regulating the activity of the promoter region [34]. The presence of the c.905G>A SNP in the promoter region of the adenosine monophosphate deaminase 1 (*AMPD1*) gene in chickens was found to alter the ability to bind transcription factors, and the rate-limiting step of the purine nucleotide cycle was enhanced, mainly maintaining freshness of fresh chicken meat [35]. In goats, the transcription factor SP1 binds to the insulin-like growth factor 1 (*IGF1*) g.64943050 polymorphic site, regulating the proliferation of goat granulosa cells [36]. In this study, two SNPs were found in the promoter region of *PPP2R5C* and *SLC39A5* via the KASP genotyping technique, including the g.65977743C>T mutation in *PPP2R5C* and the g.50676693T>C polymorphism in *SLC39A5*. In the Yunshan black goat population, g.65977743C>T was also found to have both *TT*, TC and *CC* genotypes, and *SLC39A5* g.50676693T>C was found to have all three genotypes, *TT*, *TC* and *CC*. The results show that the two polymorphisms were moderately polymorphic, suggesting that both genes were genetically diverse in the Yunshang black goat population (0.25 < PIC < 0.5), and both were in Hardy–Weinberg equilibrium. The polymorphic information content (PIC) can reflect the level of polymorphism within a population [37]. The higher the value of polymorphism information content, the richer the genetic diversity of the population and the more favorable the genetic potential of the species for selection and selection [38]. A high value indicates that SNPs have not been strongly selected and the genes might not be influenced by selection and genetic drift. We evaluated the relationship between these two SNPs and the litter size produced. The results show that both loci exhibited a significant correlation (*p* < 0.05) with the litter size produced by black goats on clouds. These results suggest that these two SNPs may be used as candidate molecular markers for breeding highly fertile goats.

In addition, the expression in goat ovaries of different genotypes of *PPP2R5C* and *SLC39A5* was validated. The expression of both genes was significantly different between goats. To date, studies on *PPP2R5C* and *SLC39A5* have focused on human diseases and mice. PP2A controls human cell proliferation, differentiation, apoptosis and migration [39,40]. In mice, *PPP2R5C* affects the Wnt/β-catenin signaling pathway through regulation of B56 to control embryonic development [41]. *PPP2R5A* is a key regulator of embryonic stem cell proliferation and apoptosis in endometrial stromal cells of bearded sheep and the number of lambs produced is affected [42]. *SLC39A5* is a member of a family of zinc transporter proteins that play an important regulatory role in mammals [18]. In zebrafish, the role of zinc homeostasis is regulated by *SLC39A5* to provide a zinc homeostatic environment for vascular endothelial cell proliferation and migration [19]. It was shown that the BMP/TGF-β pathway is regulated by *SLC39A5*-related polymorphisms and is actively involved throughout development in mice [43]. The dynamic balance of zinc during pregnancy in mice is regulated by *SLC39A5* expression [44]. Together, these results show that *PPP2R5C* and *SLC39A5* could be studied as candidate genes associated with high-fertility goat breeding.

Transcriptional regulation is an important means of controlling cellular function [45]. Transcription factors are proteins that bind to specific DNA sequences and control the rate at which genes are transcribed to produce messenger RNA [46]. Gene expression is primarily regulated at the transcriptional level, which is mainly the result of binding of transcription factors to specific DNA loci [47]. Transcription factors play an important role in transcription initiation in eukaryotes [36]. To further investigate the effect of polymorphisms in the promoter regions of *PPP2R5C* and *SLC39A5* on goat reproductive performance, we predicted transcription factor binding sites near the variant sites and found that the *PPP2R5C* g.65977743C>T variant produced new transcription factors *SOX18*, *ZNF418* and *ZNF667*. We also identified new transcription factors *NKX2-4* and *TBX6* in the *SLC39A5* g.50676693T>C variant. This provides a basis for the exploration of the molecular mechanisms of these two loci.

## 5. Conclusions

In conclusion, SNPs detected in the promoter regions of *PPP2R5C* (g.65977743C>T) and *SLC39A5* (g.50676693T>C) were found to be significantly associated with litter size in Yunshang black goats. Moreover, both polymorphisms resulted in a change in the promoter activity of the respective genes.

## Figures and Tables

**Figure 1 animals-12-02801-f001:**
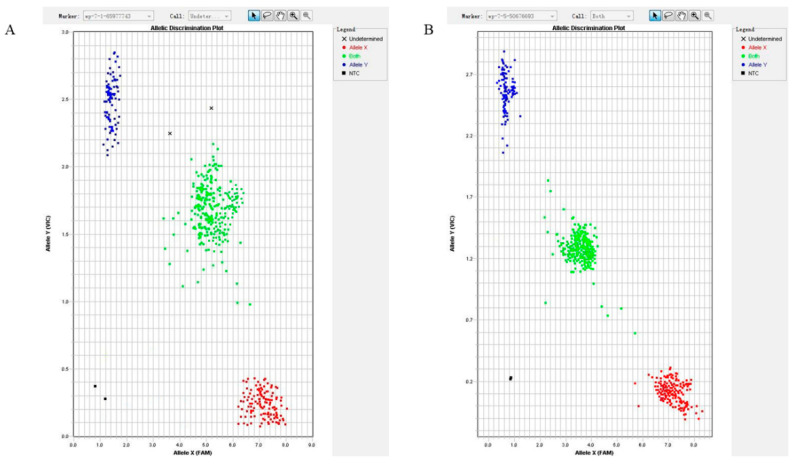
Genotyping of SNPs by KASP method. (**A**) *PPP2R5C* g.65977743C>T; and (**B**) *SLC39A5* g.50676693T>C. Note: blue color dots, green color dots and red color dots represent the individuals with *TT*, *TC* and *CC* genotype, respectively.

**Figure 2 animals-12-02801-f002:**
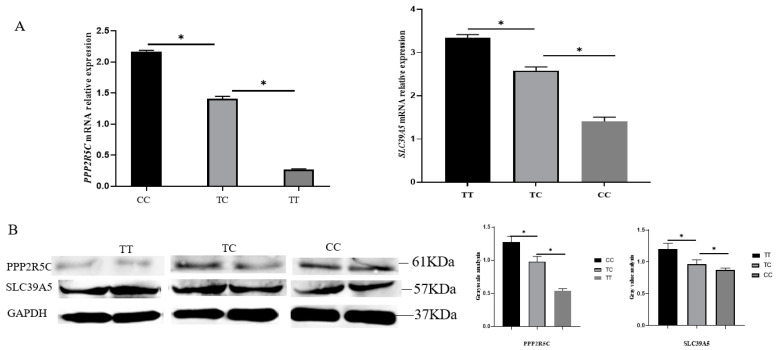
Expression of *PPP2R5C* and *SLC39A5* with different genotypes in goat ovarian tissues. (**A**) The mRNA expression of *PPP2R5C* and *SLC39A5*; (**B**) the protein expression and gray value of PPP2R5C and SLC39A5. * *p* < 0.05.

**Figure 3 animals-12-02801-f003:**
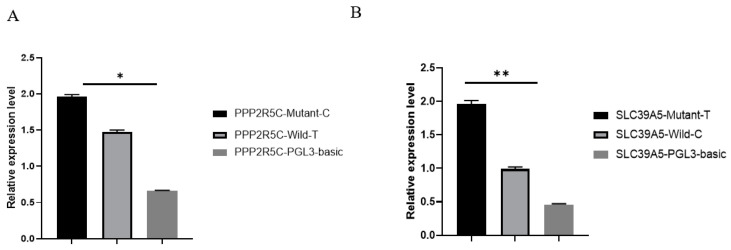
Effect of polymorphisms on the promoter activity of *PPP2R5C* and *SLC39A5.* (**A**) The transcriptional activity of the *PPP2R5C* promoter region with different haplotypes; (**B**) the transcriptional activity of the *SLC39A5* promoter region with different haplotypes. Data are expressed as the mean ± SD. * *p* < 0.05; ** *p* < 0.01.

**Figure 4 animals-12-02801-f004:**
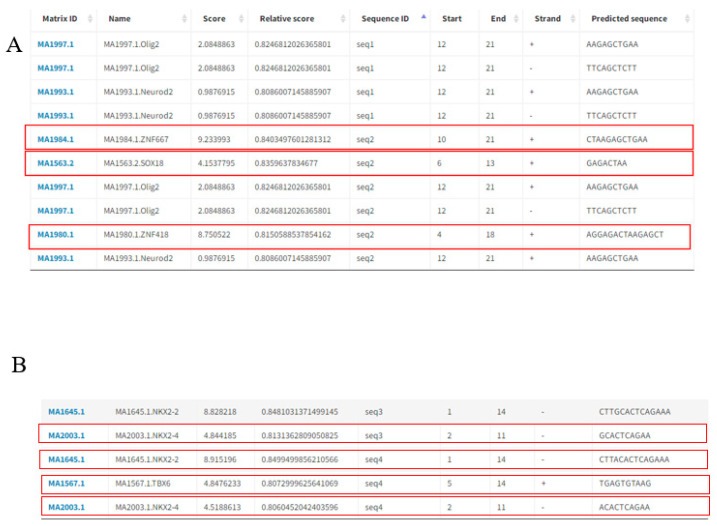
Predicted transcription factors associated with the *PPP2R5C* g.65977743C>T and *SLC39A5* g.50676693T>C polymorphisms (Red box: transcription factor that binds to the mutation site). (**A**) The *PPP2R5C* g.65977743C>T polymorphism is predicted by the transcription factor sequence seq1: GATAGGAGACTAAGAGCTGAA; seq2: GATAGGAGACCAAGAGCTGAA; (**B**) the *SLC39A5* g.50676693T>C polymorphism is predicted by the transcription factor sequence seq3. ACCCCAGTAACGCACCCAGCC; seq4: ACCCCAGTAATGCACCCAGCC.

**Table 1 animals-12-02801-t001:** Phenotypic distribution of female goats with different litter sizes.

Litter Size	Sample Size
single	67
double	204
triple	298

**Table 2 animals-12-02801-t002:** Primer sequences for the promoter regions of *PPP2R5C* and *SLC39A5*.

Gene Name	Primer Name	Sequence (5′-3′)	Length (bp)	Temperature (°C)
*SLC39A5*	*SLC39A5-1*	F: GGTGACCCTTGTCCACCTG	1226	59
		R: TACTCGGGGGACAGGCTTAGAG		
	*SLC39A5-2*	F: AGGGGACCCTGGAAGAGAC	350	59
		R: CTTCCTGGGCTTGAGACCCAC		
	*SLC39A5-3*	F: GAGCCCCAGCCCCTTCC	489	59
		R: TGATGTCAGGAAGGCAGAGCTAG		
	*SLC39A5-4*	F: CTCATGGAGGTTGTAGAAAACT	313	59
		R: GGCAGAAAATAGGCAAAGAACA		
	*SLC39A5-5*	F: GGCCTCTAGTTACTGGGTGGG	608	59
		R: GGTGGCTCCCAGTGGAGG		
* PPP2R5C *	* PPP2R5C-1 *	F: CCCAGGATACTCTGATCAGAAATGATGT	1024	59
		R: CTAGACTTCAGCGGGAAAGGCA		
	* PPP2R5C-2 *	F: TTGCCTTTCCCGCTGAAGTCT	173	59
		R: CCATTTATTAACTAACCACAGAAGGTTCCGT		
	* PPP2R5C-3 *	F: GAGTTAAGAAACATAGAAACTTTGTCATATGAAG	629	59
		R: TTCCCTCACTGTGGTCTAGGG		
	* PPP2R5C-4 *	F: CCTGGCACAGAGACTTTTCTTA	364	59
		R: TATACCACCTGGCTGTGAAAAT		
	* PPP2R5C-5 *	F: CACCACCTCATCTGTGGCT	350	59
		R: TTCAGATTACCCGAAGGGCAAGAAC		
	* PPP2R5C-6 *	F: CCAGGCTTGGCTCGTCC	425	59
		R: AACTTGATCTAAGATGTACAGATGGGAGGT		

**Table 3 animals-12-02801-t003:** Primer information for SNP genotyping.

Gene Name	Primer (5′-3′)	Primer_AlleleFAM (5′-3′)	Primer_AlleleHEX (5′-3′)
*PPP2R5C* g.65977743C>T	AATGATGTCCTTTCTTGGAGCAG	Gaaggtgaccaagttcatgct TAAAAGCTGTTATT CAGCTCTTA	Gaaggtcggagtcaacggatt TAAAAGCTGTTATT CAGCTCTTG
*SLC39A5* g.50676693T>C	ACTGTGGACTTGTTCTCTGTTCTT	Gaaggtgaccaagttcatgct TGCTGAAGGCTGG GTGCA	Gaaggtcggagtcaacggatt TGCTGAAGGCTGG GTGCG

**Table 4 animals-12-02801-t004:** RT-qPCR primers information.

Gene Name	Sequence (5′-3′)	Length (bp)	Temperature (°C)
*SLC39A5*	F: CAGCACCACTGTAGCGGTCTTC	162	59
	R: GGACTCCAGACACCAGGCTCAG		
*PPP2R5C*	F: TCAAGCGAGCACCATCAGCATC	140	59
	R: GCGAGCCTCTTCGTTCACTGTC		
*RPL19*	F: ATCGCCAATGCCAACTC	154	60
	R: CCTTTCGCTTACCTATACC		

**Table 5 animals-12-02801-t005:** Genetic characterization of Yunshang black goats.

Gene Name	SNP Position	Genotype Frequency			Allele Frequency		PIC	He	Ne	χ^2^
		*TT*	*TC*	*CC*	T	C				
*PPP2R5C*	g.65977743C>T	0.19	0.39	0.42	0.38	0.62	0.37	0.47	1.89	0.01
*SLC39A5*	g.50676693T>C	0.31	0.45	0.23	0.54	0.46	0.37	0.49	1.99	0.03

**Table 6 animals-12-02801-t006:** Analysis of associations between polymorphisms and litter size in Yunshang black goats.

Gene Name	Loci	Genotype	1st Parity Litter Size	2st Parity Litter Size	3st Parity Litter Size	Average Litter Size
		*CC*	1.96 ± 0.620	2.28 ± 0.135	2.38 ± 0.096 ^a^	2.21 ± 0.070
*PPP2R5C*	g.65977743C>T	*TC*	1.95 ± 0.070	2.18 ± 0.069	2.30 ± 0.054 ^b^	2.16 ± 0.036
		*TT*	1.93 ± 0.048	2.15 ± 0.138	2.28 ± 0.072 ^b^	2.14 ± 0.051
		*TT*	1.97 ± 0.085	2.25 ± 0.094	2.47 ± 0.079 ^a^	2.21 ± 0.053
*SLC39A5*	g.50676693T>C	*TC*	1.92 ± 0.0066	2.21 ± 0.084	2.23 ± 0.053 ^a^	2.14 ± 0.037
		*CC*	1.90 ± 0.0495	2.17 ± 0.113	2.16 ± 0.104 ^b^	2.09 ± 0.028

Note: Different letters represent the significant difference (*p* < 0.05).

## Data Availability

Data are available on request.

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
