# Peer review of "Analysis of the Association of Two SNPs in the Promoter Regions of the PPP2R5C and SLC39A5 Genes with Litter Size in Yunshang Black Goats"

_animals, 2022, doi:10.3390/ani12202801_

Round 1

Reviewer 1 Report

The manuscript from Wang et al detected the association of two SNPs in the promoter of PPP2R5C (g.65977743 T>C) and SLC39A5 (g.50676693 T>C) with total number born (TNB) trait in Yunshang black goats (n=369), and found that the TT and TT are dominant genotypes which are significantly associated with the kidding number of the third litters, and also induce the transcription of two host genes, might through influencing the binding of transcription factors, such as SOX18 and ZNF418. The manuscript is interesting. However, it can not be accepted in the present form until the concerns should be well-explained.

1. In this study, association analysis was performed to detect the effects of two SNPs polymorphisms on the kidding number traits of Yunshang block goats. However, authors found that SNPs were significantly associated with the 3rd parity kidding number only based on the record of 98 ewes (table 6). Nowadays, it is generally believed that the records from more than 300 individuals are more representative. Thus, I strongly recommended to increase the number of individuals with 3rd parity record and re-analyze again to increase the accuracy.

2. In line 255, author detected the expression of two host genes in the ovarian tissue which I think it is inappropriate, because there are several different cell types in the ovary tissue. Why did they choose or focus on a certain cell type as the main research object, such as theca cells, granulosa cells, or oocytes?

3. In the Fig.2B, the protein bands are over-clipped, for instance, the SLC39A5 protein bands can hardly see the band shape, it is necessary to re-editing the images. Besides, I notice that PPP2R5C images in the Fig.2B are not consistent with the original image provided in the supplementary files. It is strongly recommended that the author should indicate the selection area with dotted boxes in the original images provided in the supplementary files.

4. In the last paragraph of discussion section, the authors discussed in detail about two transcription factors which is unnecessary, because the discussions should be based on experimental results.

5. It is generally believed that the allele with high frequency is considered as the wild-type allele, so to PPP2R5C, T should be wild-type allele and C was mutant, which is different from the described in the Abstract section.

6. Several incomprehensible, incorrectly described long sentences, missing and misspelled words ? For instance, line 63-64, 67-68, 93-94, there [were] two SNPs in line 27, SOX186 in line 38, detected in line 90, pop-ulation in line 201, av-erage in line 214, 2nd and 3rd in table 6 and so on. The authors should carefully check the manuscript throughout and revise all the mistakes. It is recommended to advice an English-native expert for language editing. 

Author Response

animals-1955328

Title: Analysis of the associationof two SNPs in the promoter regions of the PPP2R5C and SLC39A5 genes with litter size in Yushang black goats

Author(s): Peng Wang, Wentao Li, Ziyi Liu, Xiaoyun He, Rong Lan, Yufang Liu and Mingxing Chu

Thank you very much for evaluating our work! We have tried our best to improve the manuscript according to the reviewer further comments.

Comments:

Reviewer 1

  1. In this study, association analysis was performed to detect the effects of two SNPs polymorphisms on the kidding number traits of Yunshang block goats. However, authors found that SNPs were significantly associated with the 3rd parity kidding number only based on the record of 98 ewes (table 6). Nowadays, it is generally believed that the records from more than 300 individuals are more representative. Thus, I strongly recommended to increase the number of individuals with 3rd parity record and re-analyze again to increase the accuracy.

Response: Thank you very much! We have added 200 Yunshang black goats with recorded the kidding number of the third parity, typed them using KASP and reanalyzed the data on gene frequencies and associations with kidding number (line23-230).

  1. In line 255, author detected the expression of two host genes in the ovarian tissue which I think it is inappropriate, because there are several different cell types in the ovary tissue. Why did they choose or focus on a certain cell type as the main research object, such as theca cells, granulosa cells, or oocytes?

Response: Thank you very much! In a previous study by our team, we found that PPP2R5C and SLC39A5 were involved in ovarian function and metabolism. However, it was not explored in which specific types of cells these two genes acted, so we used ovarian tissues from Yunshang black goat ground into powder for RNA extraction for subsequent molecular experiments.

References:

Tao L, He XY, Jiang YT, Lan R, Li M, Li ZM, Yang WF, Hong QH, Chu MX. Combined approaches to reveal genes associated with litter size in Yunshang black goats. Anim Genet. 2020 Dec;51(6):924-934. doi: 10.1111/age.12999. Epub 2020 Sep 28. PMID: 32986880.

  1. In the Fig.2B, the protein bands are over-clipped, for instance, the SLC39A5 protein bands can hardly see the band shape, it is necessary to re-editing the images. Besides, I notice that PPP2R5C images in the Fig.2B are not consistent with the original image provided in the supplementary files. It is strongly recommended that the author should indicate the selection area with dotted boxes in the original images provided in the supplementary files.

Response: Thank you very much! We have revised in the new version. We have re-edited and updated the images and marked them with underlines in the original image.

  1. In the last paragraph of discussion section, the authors discussed in detail about two transcription factors which is unnecessary, because the discussions should be based on experimental results.

Response: Thank you very much! We have revised in the new version.

  1. It is generally believed that the allele with high frequency is considered as the wild-type allele, so to PPP2R5C, T should be wild-type allele and C was mutant, which is different from the described in the Abstract section.

Response: Thank you very much! We have revised in the new version.

  1. Several incomprehensible, incorrectly described long sentences, missing and misspelled words? For instance, line 63-64, 67-68, 93-94, “there [were] two SNPs” in line 27, “SOX186” in line 38, “detected” in line 90, “pop-ulation” in line 201, “av-erage” in line 214, “2nd” and “3rd” in table 6 and so on. The authors should carefully check the manuscript throughout and revise all the mistakes. It is recommended to advice an English-native expert for language editing.

Response: Thank you very much! We have polished the language by the AJE company. Please see the attachment for the license display.

Reviewer 2 Report

Dear authors,

I will recommend your manuscript for publication, but I has some questions for edition.

L2 – add “…SLC39A5 genes with…”

L89 – If you are using KASP – describe this method with reference.

L110 – why you are use 10 ml of blood? For genotyping 200 mkl is enough.

L133 – “Primer Primer5” – check

L146 – not clear, how many samples were sequenced and KASP genotyped.

L249 – In Materials and Methods you are sign significant difference * and **. Why in table you are using a and b?

L263 – Invalid gel illustration – not clear full picture with ladder lines

L289 – In figure are presents only one variant of binding site, better is show two variants for easy comparison.

L307 – not clear – you were found new SNP or they were described previously?

L382 and other – you are not using correlation analysis, change it to “associated”

Regards,

Author Response

animals-1955328

Title: Analysis of the associationof two SNPs in the promoter regions of the PPP2R5C and SLC39A5 genes with litter size in Yushang black goats

Author(s): Peng Wang, Wentao Li, Ziyi Liu, Xiaoyun He, Rong Lan, Yufang Liu and Mingxing Chu

Thank you very much for evaluating our work! We have tried our best to improve the manuscript according to the reviewer further comments.

Reviewer 2

L2 – add “…SLC39A5 genes with…”

Response: Thank you very much! We have revised in the new version (Line 2).

L89 – If you are using KASP – describe this method with reference.

Response: Thank you very much! We have revised in the new version (Line 83-88).

L110 – why you are use 10 ml of blood? For genotyping 200 mkl is enough.

Response: Thank you very much! We have revised in the new version. We took a total of 10 mL of blood per goat during the sampling process. This experiment, we used only 300 µL of blood for DNA extraction (Line 108).

L133 – “Primer Primer5” – check

Response: Thank you very much! We have revised in the new version (Line 128).

L146 – not clear, how many samples were sequenced and KASP genotyped.

Response: Thank you very much! We have revised in the new version (Line 151).

L249 – In Materials and Methods you are sign significant difference * and **. Why in table you are using a and b?

Response: Thank you very much! In the multiple analysis of variance, it is more intuitive to use letters to indicate the variability. The same letter indicated that there is no significant difference between the two groups.

L263 – Invalid gel illustration – not clear full picture with ladder lines

Response: Thank you very much! We have revised in the new version (Line 145).

L289 – In figure are presents only one variant of binding site, better is show two variants for easy comparison.

Response: Thank you very much! We have added the figure in the new version (Figure 4).

L307 – not clear – you were found new SNP or they were described previously?

Response: Thank you very much! We have revised in the new version (Line 226). The SNP loci in this study have been identified in the results of the resequencing of  Yunshang black goats by our team earlier (Tao et al, 2020).

L382 and other – you are not using correlation analysis, change it to “associated”

Response: Thank you very much! We have revised in the new version (Line ).

Reviewer 3 Report

Presented manuscript entitled "Association analysis of two SNPs in the promoter regions of PPP2R5C and SLC39A5 with kidding number trait in Yunshang black goats" might be published in Animals journal but needs lots additional work.

Number of animals in experiment, applied techniques, statistical and bioinformatics analysis are  well chosen, however way of results presentation must be improved.

WHOLE TEXT MUST BE ALSO CORRECTED BY NATIVE SPEAKER!

Below are my comments to presented text:

·        line 3 and whole text - replace "kidding number" by "litter size"

·        line 13 and whole text - replace "mutation" by "polymorphism"; mutation is where allele frequency is below 0.01

·        line 16 and whole text - replace "third litters" by "third parity"

·        line 16 - "Further analysis revealed that both mutations were in the promoter region of PPP2R5C and SLC39A5, respectively" - it was known before association analysis because authors searched for polymorphism in this region

·        line 18 and whole text incl. tables - italicize genotypes - e.g. CC

·        line 21 - "...are the potential molecular markers..." wording is too far reaching

·        line 25 - SNP when single, SNPs when more than 1 polymorphism

·        lines 35-36 and whole text - do not use "mutant" word, it will be better SLC39A5-C variant or allele

·        line 41 and whole text - use italics when mention gene name; do not use italics for proteins name e.g. line 52,58

·        line 46 - goat is not a breed

·        line 50 - replace "marks" by "markers"

·        line 52 - enzyme is not located on goat chromosome 21, just gene which encode its

·        lines 55 and 85, 70 and 86 - duplicated information

·        line 66, 67, 296, 297, 298, 317 and whole text - give abbreviations

·        line 82 vs. 88 -  sentences contradict each other

·        AT THE END OF INTRODUCTION CLEAR AIM OF STUDY SHOULD BE GIVEN!

·        line 111 - "anticoagulated with EDTA-K2 anticoagulant" replace by "treated with K2EDTA anticoagulant"

·        line 112 vs. 255 - i am not sure how 6 animals for RNA isolation were chosen; first" three low-prolificacy and three high-prolificacy individuals" then " individuals with different genotypes in high- and low-prolificacy goat ovarian tissues" - explain it

Whether the authors were exceptionally lucky that among 6 animals with different prolificacy found each genotype of both genes?

·        Table 1 - Litter size - single, double, triple

·        line 135 - "Based on Sanger sequencing results were analyzed by SeqMan software. " delete "based on"

·        Line 136 - "Generation sequencing was performed to determine mutation site information of PPP2R5C and SLC39A5 based on the sequencing results, followed by Kasp genotyping." rewrite this sentence; what is "Generation sequencing"?

·        Decide to use KASP or kasp in whole text.

·        line 140 - "upstream and downstream" better "forward i reverse" as in the Table 2.

·        Table 2 - i am not sure if is possible to sequencing fragment longer than 800bp (maybe 1000bp with long capillaries) by Sanger method; Why authors used 5-6 primers pairs for each gene? I think it is possible to sequencing 2000bp fragment with 3 pairs (3x800bp); even 2 pairs (2x1000bp).

·        line 147 - "Primers were designed and detected for the PPP2R5C and SLC39A5 SNPs loci using the KASP typing technique."- i think KASP is for genotyping not primer design and detect; what is primer detection?

·        Table 3 - In KASP method 3 primers are used. Column 2 and 5 are duplicated.

·        line 201 - population

·        line 208 - delete "In the formula" and put "where:"

·        line 219 - delete sentence " To further verify the presence of polymorphic mutations in Yunshang black goats, blood samples from 369 Yunshang black goats were used to KASP genotyping."

·        Authors should clear state that sequencing allowed for discover 2 SNPs. KASP method, however were used for SNPs genotyping what is shown on Fig. 1.

·        Figure 1. description will be better if change for:

"Figure 1. Genotyping of SNPs by KASP method

PPP2R5C g.65977743T>C; (B) and SLC39A5 g.50676693T>C

Note: blue color dots, green color dots and red color dots represent the individuals with TT , TC  and CC genotype, respectively."

·        lines 243-246 - duplicated information

·        line 239 - P>0.05?

·        Tab. 5 - why PIC, He, Ne, P are italicized? Do it with genotype names. Replace "mutation sites" on "SNP position"

·        Tab 6. In upper row "Litter size" should be put, below only "1st parity, 2nd parity, 3rd parity"

·        line 255 - genotypes

·        line 283 - 287 - Do allele C abolish TFBS in case of both genes? It should be mentioned. Allele C may also create site for other TF than SOX18 and ZNF418.

·        line 296 - locus was, loci were

·        DISCUSSION IS WEAKEST POINT OF THE PAPER AND SHOULD BE IMPROVED!

Authors should discuss own results with other authors e.g. other genes associated with litter size in goats, genes analyzed in Yunshang black goats in different aspects, association analysis of PPP2R5C and SLC39A5 in other species.

Many information are repeated (e.g. lines 300-309, 321-328, 339-348).

·        line 299 - The kidding number trait produced?

·        line 310 - However? Authors did not mention that earlier given SNPs were in coding regions (294-298)

·        line 314 - SNP mutations?

·        line 317 - "were identified in fresh chicken" what?

·        line 329 - homogenity?

·        LAST SENTENCE IN CONCLUSION IS TOO FAR REACHING.

Author Response

animals-1955328

Title: Analysis of the associationof two SNPs in the promoter regions of the PPP2R5C and SLC39A5 genes with litter size in Yushang black goats

Author(s): Peng Wang, Wentao Li, Ziyi Liu, Xiaoyun He, Rong Lan, Yufang Liu and Mingxing Chu

Thank you very much for evaluating our work! We have tried our best to improve the manuscript according to the reviewer further comments.

Reviewer 3

Number of animals in experiment, applied techniques, statistical and bioinformatics analysis are  well chosen, however way of results presentation must be improved.

Response: Thank you very much! We have improved in the new version.

WHOLE TEXT MUST BE ALSO CORRECTED BY NATIVE SPEAKER!

Response: Thank you very much! We have polished the language by the AJE company. Please see the attachment for the license display.

  • line 3 and whole text - replace "kidding number" by "litter size"

Response: Thank you very much! We have revised in the new version.

  • line 13 and whole text - replace "mutation" by "polymorphism"; mutation is where allele frequency is below 0.01

Response: Thank you very much! We have revised in the new version.

  • line 16 and whole text - replace "third litters" by "third parity"

Response: Thank you very much! We have revised in the new version.

  • line 16 - "Further analysis revealed that both mutations were in the promoter region of PPP2R5C and SLC39A5, respectively" - it was known before association analysis because authors searched for polymorphism in this region

Response: Thank you very much! We have revised in the new version (line 16).

  • line 18 and whole text incl. tables - italicize genotypes - e.g. CC

Response: Thank you very much! We have revised in the new version.

  • line 21 - "...are the potential molecular markers..." wording is too far reaching

Response: Thank you very much! We have revised in the new version (line ?).

  • line 25 - SNP when single, SNPs when more than 1 polymorphism

Response: Thank you very much! We have revised in the new version.

  • lines 35-36 and whole text - do not use "mutant" word, it will be better SLC39A5-C variant or allele

Response: Thank you very much! We have revised in the new version.

  • line 41 and whole text - use italics when mention gene name; do not use italics for proteins name e.g. line 52,58

Response: Thank you very much! We have revised in the new version. (line 56, 61)

  • line 46 - goat is not a breed

Response: Thank you very much! We have revised in the new version (line 50).

  • line 50 - replace "marks" by "markers"

Response: Thank you very much! We have revised in the new version (line 54).

  • line 52 - enzyme is not located on goat chromosome 21, just gene which encode its

Response: Thank you very much! We have revised in the new version (line 56-58).

  • lines 55 and 85, 70 and 86 - duplicated information

Response: Thank you very much! We have revised in the new version.

  • line 66, 67, 296, 297, 298, 317 and whole text - give abbreviations

Response: Thank you very much! We have revised in the new version (line 70, 72, 339, 341).

  • line 82 vs. 88 - sentences contradict each other

Response: Thank you very much! We have revised in the new version (line 88-96).

  • AT THE END OF INTRODUCTION CLEAR AIM OF STUDY SHOULD BE GIVEN!

Response: Thank you very much! We have revised in the new version (line 99-105).

  • line 111 - "anticoagulated with EDTA-K2 anticoagulant" replace by "treated with K2EDTA anticoagulant"

Response: Thank you very much! We have revised in the new version (line 121).

  • line 112 vs. 255 - i am not sure how 6 animals for RNA isolation were chosen; first" three low-prolificacy and three high-prolificacy individuals" then " individuals with different genotypes in high- and low-prolificacy goat ovarian tissues" - explain it

Response: Thank you very much! We have redescription in the new version (line 116, 283). Individuals with different genotypes were selected for RNA extraction and gene expression was measured using RT-qPCR in three individuals from each group.

Whether the authors were exceptionally lucky that among 6 animals with different prolificacy found each genotype of both genes?

Response: Thank you very much! We have phrased this incorrectly and have revised in the new version (line117-125). Rather than using six prolific goats to study mutations in these two genes, three individuals of each genotype were selected for detecting the expression of these two genes based on sequencing results.

  • Table 1 - Litter size - single, double, triple

Response: Thank you very much! We have revised in the new version (Table 1).

  • line 135 - "Based on Sanger sequencing results were analyzed by SeqMan software. " delete "based on"

Response: Thank you very much! We have revised in the new version (line 145).

  • Line 136 - "Generation sequencing was performed to determine mutation site information of PPP2R5C and SLC39A5 based on the sequencing results, followed by Kasp genotyping." rewrite this sentence; what is "Generation sequencing"?

Response: Thank you very much! We have revised in the new version (line 141-151).

  • Decide to use KASP or kasp in whole text.

Response: Thank you very much! We have revised in the new version.

  • line 140 - "upstream and downstream" better "forward i reverse" as in the Table 2.

Response: Thank you very much! We have revised in the new version (line 152).

  • Table 2 - i am not sure if is possible to sequencing fragment longer than 800bp (maybe 1000bp with long capillaries) by Sanger method; Why authors used 5-6 primers pairs for each gene? I think it is possible to sequencing 2000bp fragment with 3 pairs (3x800bp); even 2 pairs (2x1000bp).

Response: Thank you very much! Due to the limitations of primer design and technical problems of the sequencing company, we determined the primer sequences in the table after several trials.

  • line 147 - "Primers were designed and detected for the PPP2R5C and SLC39A5 SNPs loci using the KASP typing technique."- i think KASP is for genotyping not primer design and detect; what is primer detection?

Response: Thank you very much! We have revised in the new version (line 162).

  • Table 3 - In KASP method 3 primers are used. Column 2 and 5 are duplicated.

Response: Thank you very much! We have revised in the new version.

  • line 201 - population

Response: Thank you very much! We have revised in the new version (line 220).

  • line 208 - delete "In the formula" and put "where:"

Response: Thank you very much! We have revised in the new version (line 227).

  • line 219 - delete sentence " To further verify the presence of polymorphic mutations in Yunshang black goats, blood samples from 369 Yunshang black goats were used to KASP genotyping."

Response: Thank you very much! We have revised in the new version (line 237).

  • Authors should clear state that sequencing allowed for discover 2 SNPs. KASP method, however were used for SNPs genotyping what is shown on Fig. 1.

Response: Thank you very much! We have revised in the new version.

  • Figure 1. description will be better if change for:

"Figure 1. Genotyping of SNPs by KASP method PPP2R5C g.65977743T>C; (B) and SLC39A5 g.50676693T>C Note: blue color dots, green color dots and red color dots represent the individuals with TT , TC  and CC genotype, respectively."

Response: Thank you very much! We have revised in the new version (Figure 1).

  • lines 243-246 - duplicated information

Response: Thank you very much! We have revised in the new version (line 272-281).

  • line 239 - P>0.05?

Response: Thank you very much! We have revised in the new version (line 247).

  • Tab. 5 - why PIC, He, Ne, P are italicized? Do it with genotype names. Replace "mutation sites" on "SNP position"

Response: Thank you very much! We have revised in the new version (Table 5).

  • Tab 6. In upper row "Litter size" should be put, below only "1st parity, 2nd parity, 3rd parity"

Response: Thank you very much! We have revised in the new version (Table 6).

  • line 255 - genotypes

Response: Thank you very much! We have revised in the new version (line 290).

  • line 283 - 287 - Do allele C abolish TFBS in case of both genes? It should be mentioned. Allele C may also create site for other TF than SOX18 and ZNF418.

Response: Thank you very much! We have revised in the new version (line 333).

  • line 296 - locus was, loci were

Response: Thank you very much! We have revised in the new version (line 344).

  • DISCUSSION IS WEAKEST POINT OF THE PAPER AND SHOULD BE IMPROVED!

Response: Thank you very much! We have revised in the new version.

Authors should discuss own results with other authors e.g. other genes associated with litter size in goats, genes analyzed in Yunshang black goats in different aspects, association analysis of PPP2R5C and SLC39A5 in other species.

Response: Thank you very much! We have revised in the new version.

Many information are repeated (e.g. lines 300-309, 321-328, 339-348).

Response: Thank you very much! We have revised in the new version (line 353-355, 367-375, 389-407).

  • line 299 - The kidding number trait produced?

Response: Thank you very much! We have revised in the new version (line 348).

  • line 310 - However? Authors did not mention that earlier given SNPs were in coding regions (294-298)

Response: Thank you very much! We have revised in the new version (line 357).

  • line 314 - SNP mutations?

Response: Thank you very much! We have revised in the new version (line 360).

  • line 317 - "were identified in fresh chicken" what?

Response: Thank you very much! We have revised in the new version (line 361-364).

  • line 329 - homogenity?

Response: Thank you very much! We have revised in the new version (line 376).

  • LAST SENTENCE IN CONCLUSION IS TOO FAR REACHING.

Response: Thank you very much! We have revised in the new version (line 424-427).

Round 2

Reviewer 1 Report

None.